# Effects of Different Intensities of Endurance Training on Neurotrophin Levels and Functional and Cognitive Outcomes in Post-Ischaemic Stroke Adults: A Randomised Clinical Trial

**DOI:** 10.3390/ijms26062810

**Published:** 2025-03-20

**Authors:** Sara Górna, Tomasz Podgórski, Paweł Kleka, Katarzyna Domaszewska

**Affiliations:** 1Department of Physiology, Poznan University of Physical Education, 61-871 Poznań, Poland; domaszewska@awf.poznan.pl; 2Department of Biochemistry, Poznan University of Physical Education, 61-871 Poznań, Poland; podgorski@awf.poznan.pl; 3Department of Psychology and Cognitive Science, Adam Mickiewicz University, 60-568 Poznań, Poland; kleka@o365.amu.edu.pl

**Keywords:** motor learning, neurological rehabilitation, physical therapy, functional status, ischaemic stroke, neurotrophins

## Abstract

This study aimed to examine the effects of different intensities of endurance training combined with standard neurorehabilitation on selected blood biomarkers and physical outcomes of post-stroke individuals. We randomised patients with first-episode ischaemic stroke to an experimental group that received 4 × 45 min sessions of moderate-intensity continuous training (MICT) each week and 2 × 45 min of standard rehabilitation each day or to a control group that received 4 × 45 min sessions of low-intensity continuous training (LICT) each week and 2 × 45 min of standard rehabilitation each day. We measured the following outcomes at baseline and 3 weeks after the intervention: aerobic capacity; cognitive and motor function; and blood levels of brain-derived neurotrophic factor (BDNF), glial cell line–derived neurotrophic factor (GDNF), vascular endothelial growth factor A (VEGF-A), insulin-like growth factor-1 (IGF-1), and irisin. We included 52 patients with a mean age of 66.1 ± 8.0 years. After 3 weeks of rehabilitation, there was a clinically significant improvement in the Rivermead Motor Assessment—arm score in the MICT group. The study showed that after 3 weeks, an intervention combining MICT with standard neurorehabilitation was significantly more beneficial in improving aerobic capacity and arm motor function than an intervention combining LICT and standard neurorehabilitation.

## 1. Introduction

Ischaemic stroke (IS) is a medical emergency involving cerebral dysfunction that leads to impaired local disturbance of blood flow to the brain vessels and loss of central nervous system function, usually manifested as decreased cognitive and motor abilities. Based on the Global Burden of Stroke analysis, there was a substantial increase in stroke incidents from 1990 to 2019. In 2019, IS constituted 7.63 million (62.4%) of all new incident cases of stroke [1]. From 1990 to 2019, disability-adjusted life-years (DALYs) increased by more than 30% in the post-stroke adult population. In 2019, cerebrovascular incidents were the second leading cause of death and the most important cause of acquired motor disability throughout the world [1].

Neurorehabilitation of secondary post-stroke dysfunctions—including limitations in cognition, memory, sensation, motor control, and balance—leads to high public health costs [2]. Undertaking physical aerobic exercise that is tailored to each individual based on their age and physical capacity might play a key role in post-stroke rehabilitation by increasing cardiovascular and aerobic fitness, mobility, and cognitive abilities. Indeed, a multicentre study involving 32 countries identified physical activity as the most important common modifiable risk factor associated with IS [3]. The most frequently noted difficulties among IS patients in participating in future physical activity are environmental barriers (e.g., challenges in ambulating independently) and personal barriers, including a lack of motivation, a fear of failing, and a lack of sufficient knowledge of initiating and maintaining an adequate physical activity level [4]. The stroke subacute rehabilitation phase seems to be essential because maintenance or improvement in aerobic fitness may impact motor function, such as the ability to walk and perform everyday activities as independently as possible. These abilities increase independence in social life and limit long-term disability.

There is extensive evidence that blood biomarkers—including the neurotrophins brain-derived neurotrophic factor (BDNF), glial cell line-derived neurotrophic factor (GDNF), insulin-like growth factor-1 (IGF-1), irisin, and vascular endothelial growth factor A (VEGF-A)—can potentially predict recovery and forecast rehabilitation outcomes in patients with neurological diseases. Neuroplasticity is reflected by the reconstruction or regeneration of damaged neural structures. Moreover, it enhances the restoration of the sensorimotor system on the scope of motor learning, neuromotor control, and consolidation of functional patterns [5,6,7].

Growing evidence indicates that BDNF controls neuronal metabolism, ensures proper neuronal function, regulates neurotransmission, and modulates neuroplasticity. Moreover, BDNF, an adipokine, has a key role in lipid and glucose metabolism functioning. Physical activity may change the level of BDNF biosynthesis and thus modulate motor control and motor learning [8].

A study conducted on animal models of stroke reported that muscle GDNF signalling in the nervous system might play a key role in pain-related behaviours and exercise-mediated reflexes. The role of GDNF in nociception and sympathetic reflexes could be useful in rehabilitation planning for patients who are experiencing musculoskeletal ischaemic dysfunctions [9].

Irisin is an exercise-stimulated myokine that is induced from a bioactive precursor protein, namely fibronectin type III domain-containing protein 5 (FNDC5). Irisin contributes to the regulation of brain metabolism and neuroinflammation in IS patients. Moreover, it has a key role in the control of glucose and lipid processes in postural muscles and adipose tissue by changing white adipose tissue into brown adipose tissue [10,11].

IGF-1 has a multitude of cellular functions, including apoptosis, differentiation, and proliferation [12]. IGF-1 is known for its intense anabolic effect and ability to activate muscle protein synthesis. During exercise, it stimulates the transport of amino acids into large muscle groups [13]. Åberg et al. [14] showed that the level of physical activity prior to IS influences the serum IGF-1 level during the first day of the acute phase of stroke as well as its tendency to change during the early stage of the regenerative-compensatory period of stroke. Post-stroke changes in the peripheral IGF-1 level are associated with favourable stroke outcomes and improved recovery [15]. Consistently, among healthy individuals, Nindl et al. [16] observed a positive correlation between the serum IGF-1 level and aerobic muscular endurance. IGF-1 determines the brain’s neuroprotective effects and enhances neuroplasticity in IS patients [17]. This neurotrophin also helps regulate amino acids in skeletal muscle during physical activity.

VEGF is a protein that may reduce the scale of cerebral infarction and plays an important role in stimulating neurogenesis and angiogenesis [18,19]. In an animal study, early treadmill rehabilitation training efficiently promoted angiogenesis and VEGF expression and accelerated the recovery of neurological function [20]. VEGF-A supports neural cell proliferation as well as neural damage after stroke [21]. VEGF-A also acts as a nerve regeneration factor and may accelerate recovery of neurological function in the early phase of IS [22].

The regenerative-compensatory period of IS is critical for motor and cognitive recovery in patients. Even with extensive research, researchers have not yet identified a peripheral serum biomarker of neuroplasticity that may be a determinant of the rehabilitation outcomes [23].

Therefore, we aimed to evaluate the effects of 3 weeks of low- or moderate-intensity cycling training on serum levels of neurotrophic factors, cardiorespiratory fitness, and functional and cognitive outcomes in patients during the regenerative-compensatory post-stroke period. Determining which neurotrophins have a beneficial effect on neurorehabilitation could contribute to the adaptation of the therapy programme to the implementation of motor activity during the critical period of post-stroke recovery.

## 2. Results

### 2.1. Flow of the Participants Through the Trial

Figure 1 shows the design of the trial and the flow of participants through the study. A total of 120 patients were assessed for eligibility, of whom 52 fulfilled the inclusion criteria and provided their written informed consent to participate. The participants were randomly and equally assigned in a 1:1 ratio to the MICT (n = 26) and LICT (n = 26) groups. Four MICT and four LICT participants did not finish the 3 weeks of exercise training. Therefore, 22 MICT and 22 LICT participants completed 3 weeks of rehabilitation procedure and were included in the analyses.

### 2.2. Characteristics of the Participants

Table 1 shows the characteristics of the LICT and MICT groups. The mean age of participants was 66.1 ± 8.0 years. In both groups, 47.7% were men (21 participants). The baseline clinical information was similar in the MICT and LICT groups.

### 2.3. Effect of Endurance Exercise Intensity on the Motor Ability and Aerobic Capacity

At week 3, motor performance measured by the RMA had improved by 6 (95% CI 3 to 9) points in the MICT group and by 5 (95% CI 2 to 8) points in the LICT group. The median between-group difference for the change in the RMA score was 1.5 (95% CI −1.8 to 4.8) points. The CI indicates some uncertainty regarding the extent of the benefit, spanning from an arguably trivial benefit (a 1.8-point reduction in the RMA score) to a substantial benefit (a 4.8-point improvement in the RMA score) (Table 2). The clearest benefit on motor function was noted in the arm domain (RMA-a). At week 3, the RMA-a score had improved by 2 points more in the MICT group compared with the LICT group (95% CI 0 to 4 points), suggesting a potential substantial benefit.

The BI score improved in both groups: by 22.5 (95% CI 4.6 to 40.4) points in the MICT group and by 25 (95% CI 11.9 to 38.1) points in the LICT group. At week 3, the median between-group difference for the change in the BI score was 10 (95% CI −0.1 to 20.1) points (Table 2).

Aerobic capacity endurance improved in both groups but more in the MICT group. At week 3, the median between-group difference for the change in the 6MWT distance was 163.8 (95% CI 100 to 227.5) m (Table 2). The bounds of the CI indicate that the MICT group might have a substantial benefit over the LICT group.

VO_2_max improved in both groups, but more in the MICT group. At week 3, the median between-group difference for the change in VO_2_max was 6.8 (95% CI 3.7 to 9.9) mL/kg/min (Table 2). The CI indicates a significant improvement, suggesting a meaningful difference between the MICT and LICT groups.

### 2.4. Effect of Endurance Exercise Intensity on Cognitive Function

At week 3, cognitive function measured by the ACE-III had improved by 4 (95% CI 0.4 to 7.6) points in the MICT group and by 0.5 (95% CI −1.1 to 2.1) points in the LICT group. The median between-group difference for the change in the ACE-III score was 6.5 (95% CI −5.1 to 18.1) points. The CI indicates some uncertainty regarding the extent of the benefit, spanning from an arguably trivial benefit (i.e., a 5.1-point reduction in the ACE-III score) to a substantial benefit (an 18.1-point improvement in the ACE-III score). After the intervention, the median between-group difference for the change in the ACE-III A score was 3.5 (95% CI 1.5 to 5.5) points. The bounds of the CI indicate that the MICT group might have a substantial benefit over the LICT group. The clinical benefit of MICT was also notable in the ACE-III V domain. After the intervention, the ACE-III V score improved by 2.5 points more in the MICT group than in the LICT group (95% CI 0.7 to 4.3 points). The CI indicates some uncertainty in the exact amount of benefit, spanning from an arguably trivial benefit (an improvement of 0.7 points in the ACE-III V score) to a substantial benefit (a 4.3-point improvement) (Table 2).

### 2.5. Effect of the Endurance Exercise Intensity on the Level of Selected Biomarkers

At week 3, the median between-group difference for the change in the peripheral BDNF level was −2.2 (95% CI −3.7 to −0.7) ng/mL (Table 3). The limits of the CI indicate that the change was statistically significant, suggesting a decrease in the peripheral BDNF concentration in the MICT group.

At week 3, the median between-group difference for the change in the GDNF level was 0 (95% CI −0.3 to 0.4) ng/mL (Table 3). The CI indicates that the change was not statistically significant, suggesting no significant difference in peripheral GDNF levels between the groups.

At week 3, the median between-group difference for the change in the circulating IGF-1 level was 2.8 (95% CI -19.1 to 24.6) ng/mL (Table 3). The wide CI indicates substantial uncertainty in the exact extent of change, making it challenging to draw definitive conclusions about the observed difference in IGF-1 between the groups.

At week 3, the median between-group difference for the change in the circulating irisin level was 40.7 (95% CI −62.6 to 143.9) ng/mL (Table 3). The wide CI indicates substantial uncertainty in the exact amount of change, making it challenging to draw definitive conclusions about the observed difference in the peripheral irisin level between the groups.

Finally, at week 3, the median between-group difference for the change in the peripheral VEGF-A concentration was 6.1 (95% CI −5 to 17.2) ng/mL (Table 3). The lower limit of the CI indicates that the change was not statistically significant, while the upper limit indicates some uncertainty in the exact extent of change.

### 2.6. Correlation Between the Effect of Exercise and the Clinical Presentation

There were significantly negative correlations between the serum irisin level and the patient’s age (r = −0.316) at baseline (Table 4). However, there were no significant correlations between the functional status, aerobic capacity, and the BDNF, GDNF, IGF-1, irisin, and VEGF-A levels at baseline or after 3 weeks of aerobic exercise (Table 4 and Table 5).

## 3. Discussion

The American Stroke Association and the American Heart Association suggest that post-stroke patients should perform the following physical activity to prevent future stroke incidents and other cardiovascular diseases: aerobic activity for 20–60 min a day, 3–5 days per week, and strength and flexibility exercises 2–3 days a week [24]. Post-stroke individuals exhibit low energy expenditure, a short activity duration, and a low frequency of participation in physical activity [25]. The average daily walking time at moderate intensity is usually less than 10 min and occurs in short bouts [26]. It is estimated that physical activity levels in the post-stroke population are significantly lower than in healthy controls. In post-stroke survivors, on average, the number of steps taken is reduced by half [27,28]. In a systematic review and quantitative synthesis of 103 studies including 5306 post-stroke participants, Fini et al. [27] revealed that people who experienced stroke took, on average, 5535 steps per day during the subacute period of stroke and 4078 steps per day during the chronic period of stroke. Moreover, the daily sedentary time was more than 78% of daily activities in the stroke population.

Among post-stroke survivors, a decreased activity level and extended time spent in a sitting position are often reported. Inpatients who participate in stroke rehabilitation and have gained the ability to walk independently present a significantly increased amount of non-therapy activity time [29]. Barrett et al. [30] reported similar findings. During their 2-week study on inpatient stroke rehabilitation survivors, they observed that patients spent 86.9% of their waking hours sedentary. Moreover, 61.6% of the total time during physical therapy was carried out in a sitting position. The average HR increased by about 15 beats per minute during physical therapy dedicated to stroke patients. These authors also emphasised that therapy sessions are too infrequent and have too low intensity. Based on these results, the physical activity time that patients undertake during rehabilitation is below the World Health Organization’s (WHO) suggestion for moderate and vigorous physical activity [31]. The cardiorespiratory fitness of patients who have experienced a stroke is reduced by half compared with patients who have not had a stroke [32].

In our study, after 3 weeks of rehabilitation, we observed the most evident benefit on motor function based on the RMA-a score: the arms ability was improved by 2 points more in the MICT group than the LICT group (95% CI 0 to 4 points). The limits of the CI indicate that the MICT group might have a substantial benefit over the LICT group (improved by 4 points on the 0–15-point scale). Although aerobic capacity improved in both groups, it was more clinically significant in the MICT group. At week 3, the median between-group difference for the change in the 6MWT distance was 163.8 (95% CI from 100 to 227.5) m. Moreover, VO_2_max improved in both groups but more meaningfully in the MICT group (6.8 [95% CI 3.7 to 9.9] mL/kg/min).

In a randomised clinical trial involving 70 patients (mean age: 57.6 ± 9.2 years) whose first stroke occurred between 3 months and 5 years ago, the authors examined the effect of a high-intensity interval training (HIIT) intervention. The HIIT intensity was 85–95% of peak HR. During the 8 weeks of the study, the patients were also exposed to standard physical therapy care. After combining HIIT with standard care, there was an improvement in physical functioning—an increase in walking distance and balance—as well as cognitive function and executive abilities [33]. A recent systematic review of 35 randomised controlled trials showed that endurance exercise might enhance the cardiorespiratory function in stroke patients. HIIT was noted as the most effective training in improving cardiorespiratory fitness measured by peak oxygen uptake (VO_2_peak) and motor abilities determined by the 6MWT [34].

Steen Krawcyk et al. [35] conducted a randomised clinical trial of 71 patients with lacunar minor stroke, and assessed the relationship between cognition, endothelial biomarkers, and cardiorespiratory fitness in patients undergoing 12 weeks of home-based HIIT or usual care. The study group was slightly younger (63.7 ± 9.2 years) compared with the out MICT group. There were no significant changes in the median VEGF level between baseline (29.8 pg/mL) and post-intervention (30.3 pg/mL) in the study group. Moreover, after 12 weeks of home rehabilitation, HIIT did not yield a consistent effect on cardiorespiratory fitness or cognitive function compared with usual care. The authors concluded that an exercise intensity that was too low or the short daily HIIT duration (3 × 3 min with 2 min of active recovery) may have been the reason for the lack of change in the VEGF level. Based on a systematic review of 20 clinical trials investigating the effect of HIIT in cardiovascular patients, Kolmos et al. [36] showed that this training seems likely to be an advantageous time-efficient training strategy in neurorehabilitation. However, Hussain et al. [37] stated that there are no differences in the clinical utility of HIIT and MICT protocols in people experiencing cardiovascular conditions, especially stroke. The patients who participated in our study were in the early phase of the regenerative-compensatory period and had a VO_2_max in the lower limit of normal, with a median of 25 (95% CI 19 to 32) mL/kg/min in the MICT group and 20 (95% CI 15 to 24) mL/kg/min in the LICT group. At the beginning of the study, the patients participating in training on a cycle ergometer did not use the HIIT protocol.

Luo et al. [23] evaluated the relationship between the BDNF level and various parameters in patients after acute stroke. The study group consisted of 348 post-acute patients with a mean age of 67.7 ± 15.2 years, with slightly more males (56.3%) than females. The authors concluded that the serum BDNF level has a minimal predictive clinical effect on the motor status in early stroke rehabilitation. Wang et al. [38] showed that a low BDNF level was significantly correlated with poor motor skills during the subacute phase of stroke. Stanne et al. [39] proposed a BDNF cut-off of 21.8 ng/mL as an indicator for predicted long-term recovery in individuals after an acute IS. A low BDNF level has been linked with worse functional results in the future. Contrary to these results, in our study, there was no relationship between the BDNF level and functional capabilities of patients during the subacute period after stroke.

Haavik et al. [40] conducted a prospective clinical study of patients with chronic stroke, assessing changes in the BDNF and GDNF levels in patients undergoing an intensive physical therapy intervention (3 × 40 min a week for 4 weeks) and 15 min of chiropractic spinal intervention. This rehabilitation contributed to a significant decrease in the BDNF log level and a significant increase in the GDNF log level at the 8-week post-rehabilitation follow-up. In our study at week 3, the median between-group difference for the change in the peripheral BDNF level was −2.2 (95% CI −3.7 to −0.7) ng/mL, indicating a significant decrease in the peripheral BDNF level in the MICT group compared with the LICT group. In contrast, Lukkahatai et al. [41], in a scoping review revealed, that peripheral levels of BDNF increase after exercise. The inconsistent findings on BDNF changes may have been partly due to neuroendocrinological mechanisms such as hypothalamic–pituitary–adrenal axis or adrenal gland function [42,43]. The above-mentioned phenomenon ought to be clarified, and further research is needed.

Alves et al. [44] assessed the effect of 12 weeks of aerobic physical exercise in chagasic mice. The heart GDNF level was higher after training than before training. Pedroso et al. [45] investigated whether the GDNF level is associated with depressive symptoms during the acute phase of stroke. They noted a more severe manifestation of depression in patients with a lower GDNF level. Domaszewska et al. [46] studied 40 postmenopausal women with a mean age of 66.64 ± 4.20 years. Participation in Nordic walking training twice a week for 8 weeks did not alter the serum GDNF level and cognitive function measured by the Verbal Fluency Test (VTF) and the Stroop Colour and Word Interference Test. Kong et al. [47] evaluated the relationship between serum BDNF and GDNF levels, aerobic exercise, and cognitive functioning. The study group consisted of 80 fitness club members aged 27–44 years (71.25% women), and the control group comprised 80 community residents aged 31–47 years (58.75% women) who did not engage in physical activity regularly. The authors assessed cognitive performance by using the Repeatable Battery for the Assessment of Neuropsychological Status (RBANS). Visuospatial ability correlated positively with the BDNF and GDNF levels (r = 0.606 and 0.561, respectively, *p* < 0.05). The BDNF and GDNF levels were higher in the intervention group. In our study, we did not observe significant changes in the GDNF level after the 3-week intervention.

Prodjohardjono et al. [48] enrolled 56 first-episode IS patients from Indonesia. They found that a VEGF level > 519.8 pg/mL had a significant impact on post-stroke cognitive impairment (PSCI) measured by the Montreal Cognitive Assessment (MoCA)—in particular, a decrease in visuospatial and recall area in the compensatory-regenerative phase of rehabilitation. Moreover, individuals with a peripheral VEGF level ≥ 519.8 ng/mL experienced PSCI approximately 5 times more than those patients with lower peripheral VEGF concentration (odds ratio 5.05, 95% CI 1.26 to 20.32, *p* = 0.016). In another study including 171 post-IS patients (mean age 68.1 ± 10.1 years), the plasma VEGF level was increased immediately post-stroke for all IS subtypes [49]. A higher peripheral VEGF level was associated with lower dependence in the daily activity measured by modified Rankin Scale (mRS; 681 ± 40 pg/mL; mRS > 2) than in the better outcome group (496 ± 31 pg/mL; mRS ≤ 2) for the cardioembolic infarction subtype of IS. In contrast, there was no such relationship for the other subtypes. Sun et al. [50] found that early rehabilitation for acute IS patients lasting for two consecutive weeks for 45–60 min correlated with the VEGF level (r = 0.7002, *p* < 0.01). On day 15, after the IS in regularly rehabilitated patients, the VEGF level was 987.17 ± 87.25 ng/mL. Karakilic et al. [51] examined adult male Wistar rats undertaking regular mild treadmill aerobic exercise 3 days a week for 30 min a session for 6 weeks. The authors showed that this type of exercise significantly affected learning and memory. Furthermore, the VEGF level increased in the soleus and gastrocnemius type 1 (oxidative) and type 2 (glycolytic) muscle fibres as well as the hippocampus. Zhang et al. [52] evaluated 61 acute IS individuals 3 days after IS onset and observed that the plasma VEGF-A level was higher compared with the healthy group. Moreover, significant improvements in sensorimotor abilities were measured using the Fugel–Meyer Assessment (FMA). Włodarczyk et al. [53] conducted a study involving 32 patients (68.3 ± 9.1 years) at 3–4 weeks after a moderate IS who participated in a post-stroke neurorehabilitation programme comprising neurophysiological training (60 min), a psychotherapy session (15 min), and aerobic training (divided into two or three sessions of 10 min each) each day. After 3 weeks of rehabilitation, there was a 48% increase in the VEGF level. The authors also showed that changes in the peripheral VEGF level were not associated with better results in cognitive abilities as measured by the MoCA. Cognitive function improved significantly from 20.0 ± 1.9 points to 24.6 ± 1.9 points (*p* = 0.004) after 3 weeks of intensive neurorehabilitation. Our study did not observe a significant correlation between cognitive function and the VEGF-A level after rehabilitation. At week 3, the median between-group difference for the change in the peripheral VEGF-A level was 6.1 (95% CI −5 to 17.2) ng/mL. The lower limit of the CI indicates that the change was not statistically significant, while the upper limit indicates some uncertainty in the exact amount of change.

Luzum et al. [26] examined 453 patients (72.5 ± 11.3 years) who had experienced a stroke 3 months ago. Almost 70% of them adhered to the WHO recommendation for at least 150 min of moderate-intensity aerobic exercise per week; specifically, the mean weekly time was 252 min. The authors concluded that patients who experienced stroke and had cognitive dysfunctions were more likely to not follow the WHO recommendations and be inactive. Other authors have obtained similar results that cognitive impairment correlates with a decline in quality of life and reduces independence and participation in daily activities [54,55]. Barbay et al. [56] found that among 404 stroke patients (63.8 ±10.5 years old) who had experienced a stroke within 6 months, 49.5% had neurocognitive disorders. This indicates a high need for comprehensive therapy, including improvement of physical activity and cognitive functions.

An epidemiologic study demonstrated that a decrease in the serum IGF-1 level correlated most strongly with age, BMI, and metabolic syndrome. Physical activity did not influence circulating IGF-1 levels in the healthy population [57]. In post-stroke patients, Åberg [58] observed an increase in the peripheral IGF-1 level during the acute phase of IS and a decrease in the IGF-1 level 3 months after the ischaemic incident. Recent studies have reported that participating in the recommended amount of physical activity, which is one of the main modifiable IS risk factors, determines post-stroke severity and may have augmented functional abilities [59,60]. Åberg et al. [61] suggested that post-stroke dynamic changes in the IGF-1 level have relevance regarding the pace of achieving the assumed neurorehabilitation goals.

Wu et al. [62] examined 324 first-episode IS patients with a median age of 65 (IQR 57–78) years. The median irisin level was 291.2 (IQR 214.1–404.2) ng/mL 3 months post-IS. There was a negative correlation between the irisin level and the severity of the stroke assessed using the National Institutes of Health Stroke Scale (NIHSS) at 3 months after the stroke (r = −0.272, *p* < 0.001). Tu et al. [63] assessed 1530 IS patients with a median age of 66 years and found that a low serum irisin level may determine a poor functional status. In a randomised clinical trial involving 20 IS patients (55.5 ± 12.93 years old), Kang et al. [64] evaluated the effect of participation in rehabilitation involving 3 × 60 min sessions a week for 8 weeks. The whole-body exercise consisted of strength exercise (15 min), cardiovascular exercise (15 min), and team games (15 min) at 65–80% of each patient’s maximum HR. After 8 weeks, the experimental group showed an improvement in cardiorespiratory endurance measured with VO_2_peak (14.05 ± 6.47 mL/min/kg at baseline vs. 19.29 ± 7.64 mL/min/kg after 8 weeks, *p* < 0.05) and the 6MWT (176.34 ± 125.47 m at baseline vs. 229.05 ± 136.02 m after 8 weeks; *p* < 0.05). Moreover, there was a significant increase in the circulating irisin level from 6.20 ± 1.8 ng/mL to 6.77 ± 1.36 ng/mL. The peripheral irisin level is significantly related to a concomitant increase in skeletal muscle mass (r = 0.429) and a decrease in body fat mass (r = −0.771). Our study showed a significantly negative correlation between the serum irisin level and the patient’s age (r = −0.316). This trend is consistent with another recent study [65].

Comprehensive rehabilitation, including individual physiotherapy using neurophysiological methods, moderate-intensity physical activity, occupational therapy, and additional classes in speech therapy and psychology as needed, are the essential elements of the therapeutic procedure after IS. Based on our results, we conclude that this form of rehabilitation contributes to the improvement of cognitive function, in particular attention and visuospatial abilities, as well as motor functions and physical performance, and changes in the peripheral BDNF level. However, additional multicentre trials are required to verify the functions of other neurotrophins in the IS rehabilitation process.

### Implications and Limitations of the Study

Long-term observational studies with follow-up are needed to assess the broader effect of rehabilitation. Some authors have reported that the BDNF genotype may impact the outcome of a rehabilitation intervention and the cognitive and functional status of the stroke population. Hence, it is necessary to evaluate the effect of BDNF gene polymorphisms on different physical activity intensities and functional abilities. Moreover, the sample size in our study is quite limited. Additional trials with larger samples are needed to clarify the appropriate duration and intensity of endurance interventions for improving both motor and cognitive function in post-stroke individuals. A proper rehabilitation strategy may help widen the therapeutic window in the critical period of motor and cognitive recovery in stroke patients.

## 4. Materials and Methods

### 4.1. Study Participants

Recruitment and data collection took place between June and November of 2021 at the Department of Neurological Rehabilitation of the District Hospital in Śrem, Greater Poland Voivodeship, Poland. The patients who met the inclusion criteria were assessed by a team consisting of a physician specialist in internal medicine and medical rehabilitation, a neurologist, a physiotherapist, and a neuro-logopedist. The study was approved by the Bioethics Committee of Poznan Medical University (permission no. 378/21) and confirmed to the tenets of the Good Clinical Practice Guidelines and Declaration of Helsinki. Moreover, the study was registered at ClinicalTrials. gov under the identifier number NCT06824116.

The sample size was calculated by using G*Power 3.1.7. Based on a two-tailed 5% level of significance and a power of 80%, a minimum sample size of 30 participants would be appropriate to detect a significant minimal clinically important difference between the groups. Considering a dropout rate of 20%, an estimated 52 participants were required to achieve a total of 44 participants.

The inclusion criteria were age 21–75 years, first episode of IS confirmed by magnetic resonance imaging (MRI) or computed tomography (CT), able to walk for 4 m at a self-selected speed independently or with an assistive device (as needed), be in a stable clinical condition, able to communicate with investigators, and provided written informed consent to participate in the study. The exclusion criteria comprised aphasia, an unstable cardiac status (e.g., evidence of significant arrhythmia or myocardial ischaemia), lower extremity claudication, weight-bearing pain > 4/10, lower extremity spasticity (i.e., an Ashworth Scale score > 2), other neurological conditions in addition to stroke, chronic degenerative or inflammatory diagnoses, malignancies, and visuospatial neglect. The participants were informed that they were participating in a study to improve their functional status, but they were not provided with information on the superiority of a particular approach.

### 4.2. Physical Rehabilitation Programme

The participants were randomly assigned to one of two groups: moderate-intensity continuous training (MICT) on a bicycle ergometer and standard neurorehabilitation or low-intensity continuous training (LICT) on a bicycle ergometer and standard neurorehabilitation.

Training on a bicycle ergometer was performed four times a week for 3 weeks (12 training sessions in total). Endurance training was performed in the morning, at least two h after eating a meal. Each training session started with 5–10 min of warm-up and finished with 5–10 min of stretching as cool down. In both groups, the participants rode a cycle ergometer for 30 min at a targeted cadence of 60 revolutions per minute (RPM). The exercise intensity corresponded to 50–60% of maximal oxygen uptake (VO_2_max) for the LICT group and 70–80% VO_2_max for the MICT group. In each session, the MICT participants were encouraged to reach an exercise intensity at which they were no longer able to speak comfortably. On each participant’s hemiparetic side, handlebar supports and additional attachments to the pedal were available on the cycle ergometer and used as needed. Heart rate (HR) was monitored continuously during the training sessions (Polar Electro Oy, Kernpele, Finland;). Systolic blood pressure (SBP) and diastolic blood pressure (DBP) were measured systematically from the nonparetic arm before and after each session (Omron, M3V4, Kyoto, Japan). Endurance training or rehabilitation procedures were terminated when the patients presented signs suggested by American College of Sports Medicine (ACSM) guidelines [66]. The minimum required presence at 13 training sessions on a bicycle ergometer was adopted (80%). An internal medicine specialist was present when the exercise test was performed. His task was to ensure the exercise test was conducted safely.

Each participant attended individual neurorehabilitation sessions 11 times per week (2 physical therapy sessions from Monday to Friday for 45 min each and one session on Saturday for 45 min) during the 3-week intervention. The therapy comprised rehabilitation methods such as the Bobath concept and proprioceptive neuromuscular facilitation (PNF). The exercise programme designed individually for each patient included upper and lower limb functional training and its application to functional daily activities, postural stability control exercises, muscle strengthening and stretching exercises, neuromuscular exercises, balance exercises in the sitting and standing positions, walking and stair climbing, and sensory techniques. Moreover, group occupational therapy was delivered two times per week for 45 min. If the patient required it, they also participated in 45 min individual speech therapy and/or psychotherapy daily.

### 4.3. Outcome Measures

All outcome measures were assessed at baseline, before beginning training and rehabilitation, and 3 weeks after the training and rehabilitation initiation.

#### 4.3.1. Cardiorespiratory Fitness

##### The Graded Cycling Test with Talk Test (GCT-TT)

The GCT-TT is a reliable and valid estimate to assess submaximal exercise intensity. It evaluates variations in aerobic capacity following aerobic exercise. The GCT-TT was performed on a stationary bicycle (928E-G3, Monark, Vansbro, Sweden). The trial included a 5 min warm-up at 25 W and a self-selected comfortable cadence. The first stage involved cycling for 3 min with a 25 W workload at 60 RPM. The workload was increased by 25 W every 3 min with a target cadence of 60 RPM. During the last 10–15 s of each minute, the patient answered the question ‘Are you able to speak comfortably?’ with ‘yes’, ‘unsure’, or ‘no’. When the patient could no longer speak comfortably, the test was terminated [67,68]. This test identified the exercise intensity at which the patient perceived they were able to speak comfortably without accelerated breathing and when it was no longer possible to speak comfortably due to excessive breathing. The GCT-TT determined the power output of the cycling intervention in the MICT and LICT groups during the 3 weeks of endurance training. The last workload measured in which the individual patient could speak freely and their breathing was not accelerated was used for the LICT group. The last workload during which the individual patient could not speak freely and their breathing was accelerated was used for the MICT group.

##### Biomarkers

Blood was drawn to quantify the serum levels of selected neurotrophins (BDNF, GDNF, IGF-1, irisin, and VEGF-A) at baseline and 3 weeks after commencing training and neurorehabilitation. A 4 mL blood sample was obtained from the antecubital vein of the nonparetic arm between 07:00 and 08:00 h after overnight fasting. After incubating the sample for 30 min to allow clotting, the serum was separated by centrifugation, aliquoted at 100 μL, and stored at −80 °C until analysis. The Scientific Laboratory of the Department of Physiology and Biochemisty in Poznan University of Physical Education, Poland, performed the laboratory tests, using the DuoSet human enzyme-linked immunosorbent assay (ELISA) kits from R&D Systems (Minneapolis, MN, USA). The sensitivity of the ELISA kits was 0.058 ng/mL. The intra- and inter-assay coefficients of variation were less than 10% and 12%, respectively. The samples were read using a Synergy 2 SIAFRT multi-detection microplate reader (BioTek, Winooski, VT, USA) at the manufacturer’s recommended wavelength.

##### Body Mass Index (BMI)

BMI was calculated based on height (measured in centimetres) and body weight (measured in kilogrammes), disclosed using a body composition monitor (OMRON HBF-500-E; Kyoto, Japan). The equation was body weight/height^2^.

##### BP

Baseline BP was measured at each visit after an overnight fast, following 5 min of rest, with the patient in a supine position using an automatic BP monitor (Microlife R BP A100/ MicrolifeR BP A3L Comfort, Widnau, Switzerland). SBP and DBP were measured in the unaffected arm. BP is presented as millimetres of mercury (mmHg).

#### 4.3.2. Physical Performance

##### VO_2_max

Vo_2_max was assessed using a modified version of the protocol proposed by Åstrand and Rhyming [69] protocol with the use of a stationary bicycle (928E-G3, Monark), whereas HR was monitored using the Polar A-5 pulse metre (Polar Electro Oy, Kernpele, Finland) [70]. The predicted VO_2max_ was read from a nomogram [71] or accompanying tables and multiplied by the von Dôbeln et al. [71] age correction factors.

##### 6-Minute Walk Test (6MWT)

The 6MWT assesses functional capacity during continuous walking on a 12 m ward hallway for 6 min. The test measures the distance a patient walks in 6 min. The participant may take a standing rest as many times as needed, but the time is not stopped during rests [72]. The participant’s HR was monitored during the test. Moreover, the blood oxygen level (SpO2) was checked before and after the test. The 6MWT was terminated if the ACSM criteria were noted in the participant.

##### Rivermead Motor Assessment (RMA)

Functional mobility among post-stroke patients was assessed with the RMA, a performance-based measure developed as a clinical and research tool. This later includes 38 items that cover three sections: gross function (RMA-gf), with 13 items; leg and trunk (RMA-lt), with ten items; and arm (RMA-a), with 15 items. The total RMA score ranges from 0 to 38 points. For each task, the patients receive a score of ‘1’ if the activity is performed correctly and ‘0’ if it is not. This scoring is based on Guttman scaling, which presumes that each subsequent task is of a more difficult nature. Higher scores indicate a higher degree of functional mobility. The RMA has been recommended by the Academy of Neurologic Physical Therapy of the American Physical Therapy Association’s Stroke Taskforce (StrokEDGE II) for use in inpatient rehabilitation as an instrument with good psychometric properties and clinical utility. The RMA outcome measure has good internal consistency and good sensitivity in subacute stroke level of rehabilitation [73,74].

##### Barthel Index (BI)

The BI is a scale that measures an individual’s functional outcomes. The BI consists of 10 items that assess activities of daily living, functional mobility, and gait. The score ranges from 0 to 100 points. The higher the score, the greater the individual’s ability to self-care [75]. The BI has been recommended by the Stroke Taskforce (StrokEDGE) as a research tool for stroke populations.

#### 4.3.3. Cognitive Function

Addenbrooke’s Cognitive Examination (ACE-III) is a screening test to examine cognitive functions. It comprises five domains: (1) attention (ACE-III A, score: 0–18 points); (2) memory (ACE-III M, score: 0–26 points); (3) verbal fluency (ACE-III F, score: 0–14 points); (4) language (ACE-III L, score: 0–26 points); and (5) visuospatial ability (ACE-III V, score: 0–16). The overall ACE-III score ranges from 0 to 100 points. A higher score indicates better cognitive functioning of the examined person. The test takes 15–30 min to administer. There is also a short form of the ACE-III called the Mini-Addenbrooke’s Cognitive Examination (M-ACE); it takes 5–10 min to administer, and the score ranges from 0 to 30 points. The M-ACE has high sensitivity (0.85) and specificity (0.87) and a cut-off of ≤25 points [76,77].

### 4.4. Statistical Analysis

All analyses were conducted using R software (version R 4.3.2). For measurable variables, the median and interquartile range (IQR) were calculated. Group differences were assessed based on the median with 95% confidence intervals (CIs) estimated using Student’s t-distribution, assuming degrees of freedom equal to n − 1. Furthermore, Pearson correlation coefficients were calculated to examine the relationship between neurotrophin levels and functional outcomes, with 95% CIs derived from the normal distribution.

## 5. Conclusions

MICT with standard rehabilitation seems to be an effective strategy to improve walking distance and VO_2_max during the regenerative-compensatory period of stroke rehabilitation. Although endurance training has been increasingly identified as a significant component of post-stroke cognitive and functional recovery, an exercise protocol during the subacute stage of rehabilitation that optimises the rehabilitation process has not been established. Additional clinical studies are essential to determine whether rehabilitation protocols with long-term endurance activity effectively upregulate the peripheral levels of BDNF, GDNF, IGF-1, irisin, and/or VEGF, and whether these changes are meaningfully involved with the motor and cognitive abilities after post-stroke neurorehabilitation.

## Figures and Tables

**Figure 1 ijms-26-02810-f001:**
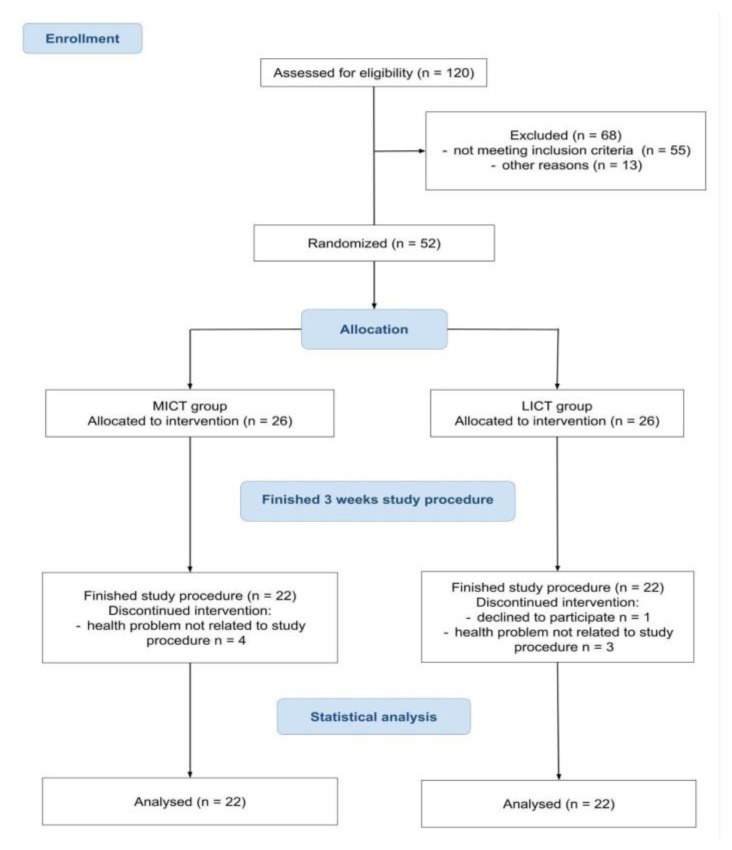
The flow of the participants through the trial.

**Table 1 ijms-26-02810-t001:** Baseline characteristics of the participants.

Characteristic	Group	
	Low-Intensity Continuous Training	Moderate-Intensity Continuous Training
	*n* = 22	*n* = 22
Age (years), mean (SD)	68.48 (5.51)	63.68 (9.38)
Gender, male (%)	11 (50.0)	10 (45.5)
Post-stroke period in weeks, mean (SD)	2.48 (1.21)	3.45 (1.63)
Hemispheric localisation of stroke, right hemisphere (%)	9 (40.9)	11 (50.0)
Height (cm), mean (SD)	166.31 (9.32)	168.45 (9.14)
Body weight (kg), mean (SD)	79.99 (17.74)	76.82 (20.38)
Body mass index, mean (SD)	28.86 (5.51)	26.89 (5.90)
Blood pressure		
Systolic (mmHg), mean (SD)	131.22 (18.41)	127.73 (16.82)
Diastolic (mmHg), mean (SD)	75.18 (11.41)	78.18 (8.68)
Risk factors, *n* (%)		
Obesity, yes (%)	10 (45.5)	8 (36.4)
Hypertension, yes (%)	20 (90.9)	17 (77.3)
Alcoholism, yes (%)	2 (9.1)	4 (18.2)
Diabetes, yes (%)	12 (54.5)	9 (40.9)
Tobacco smoking, yes (%)	6 (27.3)	9 (40.9)
Epilepsy, yes (%)	2 (9.1)	2 (9.1)
Dyslipidaemia, yes (%)	10 (45.5)	4 (18.2)
Medication, *n* (%)		
Antihypertensive/diuretic, yes (%)	20 (90.9)	19 (86.4)
Lipid-lowering, yes (%)	19 (86.4)	20 (90.9)
Antiplatelet/anticoagulant, yes (%)	21 (95.5)	21 (95.5)
Antidepressant, yes (%)	6 (27.3)	11 (50.0)
Antianxiety/sedative, yes (%)	2 (9.1)	5 (22.7)
Diabetic, yes (%)	8 (36.4)	3 (13.6)
Antipsychotic, yes (%)	3 (13.6)	2 (9.1)
Analgesic, no (%)	22 (100.0)	22 (100.0)
Insomnia, yes (%)	2 (9.1)	2 (9.1)

The data are presented as the mean (standard deviation [SD]) or the number (%).

**Table 2 ijms-26-02810-t002:** The median (interquartile range) outcomes at baseline and after the 3-week intervention, the median (interquartile range) within-group differences, and the median (95% confidence interval) between-group differences.

Outcome	Groups				Within-Group Differences		Between-Group Differences
	Baseline		After 3 Weeks		3 Weeks Minus Baseline		3 Weeks Minus Baseline
	MICT	LICT	MICT	LICT	MICT	LICT	MICT Minus LICT
	*n* = 22	*n* = 22	*n* = 22	*n* = 22	*n* = 22	*n* = 22	
Total ACE-III (points)	63 (45 to 81)	65 (46 to 84)	73 (54 to 92)	66 (44 to 89)	4 (0.4 to 7.6)	0.5 (−1.1 to 2.1)	6.5 (−5.1 to 18.1)
ACE-III A (points)	13 (11 to 15)	12 (8 to 15)	15 (12 to 18)	12 (8 to 16)	2 (1 to 3)	0 (−0.5 to 0.5)	3.5 (1.5 to 5.5)
ACE-III M (points)	14 (6 to 21)	12 (8 to 16)	16 (9 to 24)	15 (11 to 19)	1.5 (0 to 3)	0 (0 to 0)	1.5 (−2 to 5)
ACE-III F (points)	6 (2 to 10)	5 (2 to 8)	7 (4 to 10)	5 (2 to 8)	0 (−0.5 to 0.5)	0 (0 to 0)	2 (0.3 to 3.7)
ACE-III L (points)	23 (17 to 29)	23 (16 to 30)	26 (19 to 32)	24 (17 to 30)	0 (−0.5 to 0.5)	0 (0 to 0)	2 (−1.6 to 5.6)
ACE-III V (points)	13 (10 to 16)	11 (8 to 14)	14 (11 to 17)	12 (8 to 14)	0 (−0.5 to 0.5)	0 (0 to 0)	2.5 (0.7 to 4.3)
M-ACE (points)	15 (7 to 23)	16 (10 to 23)	18 (11 to 25)	16 (10 to 23)	1 (−0.5 to 2.5)	0 (−0.5 to 0.5)	1.5 (−2.1 to 5.1)
6MWT (m)	241 (106 to 377)	148 (85 to 210)	316 (190 to 442)	152 (63 to 242)	31.2 (−6.1 to 68.5)	10 (−4.5 to 24.5)	163.8 (100 to 227.5)
RMA total (points)	28 (22 to 35)	26 (16 to 36)	36 (33 to 38)	34 (28 to 40)	6 (3 to 9)	5 (2 to 8)	1.5 (−1.8 to 4.8)
RMA-gf (points)	8 (5 to 11)	7 (4 to 10)	11 (10 to 12)	11 (9 to 13)	2 (0 to 4)	2.5 (1 to 4)	0 (−1.1 to 1.1)
RMA-lt (points)	10 (9 to 11)	8 (6 to 11)	10 (10 to 10)	10 (9 to 11)	0 (−1 to 1)	1 (0 to 2)	0 (−0.5 to 0.5)
RMA-a (points)	12 (9 to 15)	10 (7 to 14)	15 (14 to 16)	13 (9 to 17)	2 (1 to 3)	2 (1 to 3)	2 (0 to 4)
BI (points)	72 (50 to 95)	55 (32 to 78)	100 (100 to 100)	90 (75 to 105)	22.5 (4.6 to 40.4)	25 (11.9 to 38.1)	10 (−0.1 to 20.1)
VO_2_max (mL/min/kg)	25 (19 to 32)	20 (15 to 24)	29 (21 to 36)	22 (17 to 27)	1.1 (−0.8 to 3)	0.9 (−0.7 to 2.5)	6.8 (3.7 to 9.9)

Abbreviations: 6MWT, 6-Minute Walking Test; ACE-III, Addenbrooke’s Cognitive Examination; ACE-III A, Addenbrooke’s Cognitive Examination—attention; ACE-III F, Addenbrooke’s Cognitive Examination—fluency; ACE-III L, Addenbrooke’s Cognitive Examination—language; ACE-III M, Addenbrooke’s Cognitive Examination—memory; ACE-III V, Addenbrooke’s Cognitive Examination—visuospatial; BI, Barthel Index; LICT, low-intensity continuous training (control group); M-ACE, Mini-Addenbrooke’s Cognitive Examination; MICT, moderate-intensity continuous training (experimental group); RMA, Rivermead Motor Assessment, RMA-a, Rivermead Motor Assessment-arm; RMA-gf, Rivermead Motor Assessment—global function; RMA-lt, Rivermead Motor Assessment—lower trunk; VO_2_max, maximal oxygen uptake.

**Table 3 ijms-26-02810-t003:** The median (interquartile range) levels of neurotrophic factors at baseline and after the 3-week intervention, the median (interquartile range) within-group differences, and the median (95% confidence interval) between-group differences.

Outcome	Groups				Within-Group Differences		Between-Group Differences
	Baseline		After 3 Weeks		3 Weeks Minus Baseline		3 Weeks Minus Baseline
	MICT	LICT	MICT	LICT	MICT	LICT	MICT Minus LICT
	*n* = 22	*n* = 22	*n* = 22	*n* = 22	*n* = 22	*n* = 22	
BDNF (ng/mL)	9 (7 to 10)	9 (7 to 11)	8 (6 to 10)	10 (8 to 12)	0.6 (−1.5 to 2.7)	1.8 (−1 to 4.6)	−2.2 (−3.7 to −0.7)
GDNF (ng/mL)	2 (1 to 2)	1 (1 to 2)	1 (1 to 2)	1 (1 to 2)	−0.2 (−0.5 to 0.1)	0 (−0.5 to 0.5)	0 (−0.3 to 0.4)
IGF-1 (ng/mL)	86 (67 to 104)	85 (53 to 118)	96 (58 to 135)	94 (55 to 133)	−0.2 (−39.5 to 39.1)	−4.1 (−31.9 to 23.7)	2.8 (−19.1 to 24.6)
Irisin (ng/mL)	509 (293 to 724)	386 (243 to 529)	478 (377 to 578)	437 (307 to 567)	−18.2 (−258.3 to 221.9)	30.5 (−148.9 to 209.9)	40.7 (−62.6 to 143.9)
VEGF-A (ng/mL)	16 (12 to 21)	18 (13 to 23)	24 (19 to 28)	18 (11 to 25)	6.9 (−0.4 to 14.2)	−1.7 (−9.7 to 6.3)	6.1 (−5 to 17.2)

Abbreviations: BDNF, brain-derived neurotrophic factor; GDNF, glial cell line–derived neurotrophic factor; IGF-1, insulin-like growth factor-1; LICT, low-intensity continuous training (control group); MICT, moderate-intensity continuous training (experimental group); VEGF-A, vascular endothelial growth factor A.

**Table 4 ijms-26-02810-t004:** Association between the patient’s cognitive and functional status, aerobic capacity, and selected biomarkers at baseline.

Variables	BDNF (ng/mL)			GDNF (ng/mL)			IGF-1 (ng/mL)			Irisin (ng/mL)			VEGF-A (ng/mL)		
	Pearson’s r	95% Upper CI	95% Lower CI	Pearson’s r	95% Upper CI	95% Lower CI	Pearson’s r	95% Upper CI	95% Lower CI	Pearson’s r	95% Upper CI	95% Lower CI	Pearson’s r	95% Upper CI	95% Lower CI
Age (years)	−0.121	0.183	−0.403	−0.080	0.222	−0.369	−0.016	0.222	−0.369	−0.316	−0.021	−0.561	0.113	0.397	−0.190
Total ACE-III (points)	−0.029	0.270	−0.324	−0.050	0.250	−0.342	−0.018	0.250	−0.342	0.073	0.362	−0.228	0.116	0.399	−0.187
ACE-III A (points)	−0.006	0.291	−0.303	−0.140	0.164	−0.419	0.005	0.164	−0.419	−0.040	0.260	−0.333	0.157	0.434	−0.147
ACE-III M (points)	−0.003	0.294	−0.300	−0.037	0.263	−0.330	−0.023	0.263	−0.330	0.132	0.413	−0.172	−0.007	0.291	−0.303
ACE-III F (points)	−0.124	0.180	−0.406	−0.052	0.248	−0.344	−0.069	0.248	−0.344	0.038	0.331	−0.262	0.006	0.302	−0.292
ACE-III L (points)	−0.046	0.255	−0.338	−0.017	0.282	−0.312	−0.046	0.282	−0.312	0.159	0.435	−0.145	0.165	0.440	−0.139
ACE-III V (points)	0.029	0.323	−0.270	0.004	0.300	−0.294	0.061	0.300	−0.294	−0.073	0.229	−0.362	0.191	0.461	−0.113
M-ACE (points)	−0.073	0.229	−0.362	−0.105	0.198	−0.390	0.007	0.198	−0.390	0.103	0.388	−0.200	0.029	0.323	−0.270
6MWT (m)	0.010	0.306	−0.288	−0.062	0.239	−0.353	0.161	0.239	−0.353	0.221	0.486	−0.081	−0.030	0.269	−0.324
RMA (points)	0.046	0.338	−0.254	0.025	0.319	−0.274	0.090	0.319	−0.274	0.045	0.338	−0.255	−0.072	0.230	−0.361
RMA-gf (points)	0.099	0.384	−0.204	−0.043	0.257	−0.336	0.111	0.257	−0.336	0.022	0.317	−0.277	−0.036	0.264	−0.329
RMA-lt (points)	0.041	0.334	−0.259	0.014	0.310	−0.284	0.091	0.310	−0.284	0.085	0.372	−0.218	0.078	0.367	−0.224
RMA-a (points)	−0.027	0.272	−0.321	0.055	0.346	−0.246	0.074	0.346	−0.246	0.033	0.327	−0.266	−0.153	0.151	−0.430
BI (points)	0.205	0.473	−0.097	−0.156	0.148	−0.432	0.065	0.148	−0.432	−0.134	0.169	−0.415	0.171	0.445	−0.133

Abbreviations: 6MWT, 6-Minute Walking Test; ACE-III, Addenbrooke’s Cognitive Examination; ACE-III A, Addenbrooke’s Cognitive Examination—attention; ACE-III F, Addenbrooke’s Cognitive Examination—fluency; ACE-III L, Addenbrooke’s Cognitive Examination—language; ACE-III M, Addenbrooke’s Cognitive Examination—memory; ACE-III V, Addenbrooke’s Cognitive Examination—visuospatial; BDNF, brain-derived neurotrophic factor; BI, Barthel Index; CI, confidence interval; GDNF, glial cell line–derived neurotrophic factor; IGF-1, insulin-like growth factor-1; LICT, low-intensity continuous training (control group); M-ACE, Mini-Addenbrooke’s Cognitive Examination; MICT, moderate-intensity continuous training (experimental group); RMA, Rivermead Motor Assessment, RMA-a, Rivermead Motor Assessment—arm; RMA-gf, Rivermead Motor Assessment—global function; RMA-lt, Rivermead Motor Assessment—lower trunk; VEGF-A, vascular endothelial growth factor A; VO_2max_, maximal oxygen uptake.

**Table 5 ijms-26-02810-t005:** Association between the patient’s cognitive and functional status, aerobic capacity, and selected biomarkers after the 3-week intervention.

Variables	BDNF (ng/mL)			GDNF (ng/mL)			IGF-1 (ng/mL)			Irisin (ng/mL)			VEGF-A (ng/mL)		
	Pearson’s r	95% Upper CI	95% Lower CI	Pearson’s r	95% Upper CI	95% Lower CI	Pearson’s r	95% Upper CI	95% Lower CI	Pearson’s r	95% Upper CI	95% Lower CI	Pearson’s r	95% Upper CI	95% Lower CI
Age (years)	−0.08	−0.37	0.22	−0.10	−0.38	0.20	−0.29	−0.55	0.02	0.06	−0.24	0.35	−0.22	−0.49	0.08
Total ACE-III (points)	0.04	−0.26	0.33	0.07	−0.23	0.36	−0.23	−0.50	0.08	−0.03	−0.33	0.27	−0.15	−0.43	0.15
ACE-III A (points)	−0.05	−0.34	0.26	−0.05	−0.34	0.25	−0.17	−0.45	0.14	0.03	−0.27	0.32	−0.20	−0.47	0.11
ACE-III M (points)	0.09	−0.21	0.38	0.04	−0.26	0.34	−0.26	−0.53	0.05	−0.05	−0.35	0.25	−0.15	−0.43	0.15
ACE-III F (points)	−0.02	−0.32	0.28	0.03	−0.27	0.32	−0.20	−0.48	0.11	0.02	−0.28	0.31	−0.13	−0.41	0.18
ACE-III L (points)	0.05	−0.25	0.35	0.16	−0.15	0.43	−0.26	−0.53	0.05	−0.09	−0.38	0.21	−0.15	−0.43	0.15
ACE-III V (points)	0.04	−0.26	0.34	0.08	−0.23	0.37	−0.06	−0.36	0.25	0.06	−0.24	0.35	−0.06	−0.35	0.24
M-ACE (points)	0.04	−0.26	0.34	−0.03	−0.32	0.27	−0.28	−0.54	0.03	−0.03	−0.33	0.27	−0.16	−0.44	0.14
6MWT (m)	0.18	−0.13	0.45	−0.16	−0.43	0.15	0.26	−0.05	0.53	0.08	−0.22	0.37	0.22	−0.08	0.49
RMA (points)	0.15	−0.16	0.42	−0.20	−0.47	0.10	0.15	−0.16	0.44	−0.10	−0.38	0.21	−0.06	−0.35	0.24
RMA-gf (points)	0.13	−0.18	0.41	−0.29	−0.54	0.01	0.17	−0.14	0.46	−0.09	−0.38	0.21	0.02	−0.27	0.32
RMA-lt (points)	0.06	−0.24	0.35	−0.23	−0.49	0.07	0.09	−0.23	0.38	−0.11	−0.39	0.20	−0.07	−0.36	0.23
RMA-a (points)	0.16	−0.14	0.44	−0.12	−0.40	0.19	0.13	−0.18	0.42	−0.08	−0.37	0.22	−0.10	−0.38	0.20
BI (points)	0.16	−0.15	0.43	−0.12	−0.40	0.18	0.19	−0.13	0.47	0.07	−0.23	0.36	0.13	−0.17	0.41

Abbreviations: 6MWT, 6-Minute Walking Test; ACE-III, Addenbrooke’s Cognitive Examination; ACE-III A, Addenbrooke’s Cognitive Examination—attention; ACE-III F, Addenbrooke’s Cognitive Examination—fluency; ACE-III L, Addenbrooke’s Cognitive Examination—language; ACE-III M, Addenbrooke’s Cognitive Examination—memory; ACE-III V, Addenbrooke’s Cognitive Examination—visuospatial; BDNF, brain-derived neurotrophic factor; BI, Barthel Index; CI, confidence interval; GDNF, glial cell line–derived neurotrophic factor; IGF-1, insulin-like growth factor-1; LICT, low-intensity continuous training (control group); M-ACE, Mini-Addenbrooke’s Cognitive Examination; MICT, moderate-intensity continuous training (experimental group); RMA, Rivermead Motor Assessment, RMA-a, Rivermead Motor Assessment—arm; RMA-gf, Rivermead Motor Assessment—global function; RMA-lt, Rivermead Motor Assessment—lower trunk; VEGF-A, vascular endothelial growth factor A; VO_2_max_,_ maximal oxygen uptake.

## Data Availability

The datasets used and analysed during the current study are available from the corresponding author on reasonable request.

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
