# Peer review of "Effects of Different Intensities of Endurance Training on Neurotrophin Levels and Functional and Cognitive Outcomes in Post-Ischaemic Stroke Adults: A Randomised Clinical Trial"

_ijms, 2025, doi:10.3390/ijms26062810_

Round 1
Reviewer 1 Report
Comments and Suggestions for Authors
The article “Effects of Different Intensities of Endurance Training on 2 Neurotrophin Levels and Functional and Cognitive Outcomes in Post-Stroke Adults: A Randomised Clinical Trial” by Gorna and coworkers reported the data obtained on selected blood neurotrophins and physical and cognitive outcomes in post-ischemic stroke patients after different intensities of endurance training combined with standard neurorehabilitation.
Two different physical rehabilitation programmes were assigned to patients: moderate-intensity continuous training (MICT) on a bicycle ergometer and standard neurorehabilitation or low-intensity continuous training (LICT) on a bicycle ergometer and standard neurorehabilitation.
The authors selected five common neurotrophins as markers of recovery and rehabilitation such as BDNF, GDNF, VEGF-A, IGF-23 and irisin, all regulating a multitude of neuronal functions, including neuron metabolism and neuroinflammation, and muscle metabolism. Blood levels of these neurotrophins were statistically correlated with cognitive function and physical performance. Interestingly, the authors found that after 3 weeks, MICT improved patients’ aerobic capacity and motor function more than LICT. The clinical beneficial effects observed after MICT are accompanied by changes in BDNF levels in peripheral blood.
In this regard, the authors suggest that determining the neurotrophins with a beneficial effect on neurorehabilitation could contribute to a personalized therapy programme aimed at improving motor activity after ischemic stroke.
Overall, the paper is interesting and well-arranged. I appreciate the clarity of the tables. However, some edits should be done:
-Line 4 of the manuscript. Authors should specify that patients are affected by ischaemic stroke and should add the word “ischaemic” in the title as follows “...and cognitive outcomes in post-ischemic stroke adults…”
-The introduction is long and redundant. In my opinion, the authors should resume the mechanism of action of the neurotrophins in a few sentences. For this reason, the introduction should be rewritten accordingly.
-I warmly suggest adding a cartoon in which the mechanism of action of the neurotrophins chosen as markers could be schematically explained.
-Lines 209-212. The sentences are redundant and should be removed. The discussion should begin with: “The American Stroke Association….”
-Paragraph titled “Limitation and Implication of the study”: In this paragraph, authors reported the hypothesis that gut microbiota may interfere with rehabilitation intervention and the levels of neurotrophins. This observation is fascinating but falls outside the theme of the present manuscript. For this reason, it should be removed.
-Line 578. The word “determined” should be replaced by the word “disclose”.
-Table 3: Authors should indicate the p-value for differences between the two experimental groups for each marker.
-The authors detected a decrease in the peripheral BDNF concentration in the MICT group compared with LICT as reported in the discussion paragraph. Contrariwise, several papers reported that the peripheral levels of BDNF increase after exercise (see for a review Lukkahatai et al., Biomedicines 2025, 13(2), 332; doi.org/10.3390/biomedicines13020332). Moreover, the authors should take into account the relationship between BDNF and the hypothalamic-pituitary-adrenal (HPA) axis that plays an important role in homeostasis maintenance (Tapia-Arancibia et al., Front Neuroendocrinol 2004; 25: 77–107). On the other hand, it is demonstrated that the adrenal gland represents a target of physical exercise (Brown et al., J Appl Physiol 2007 Dec;103(6):1979-85. doi: 10.1152/japplphysiol.00706.2007; Bartalucci et al., Histol Histopathol 2012 Jun;27(6):753-69.doi: 10.14670/HH-27.753; Hotting et al., Neural Plasticity Volume 2016, doi.org/10.1155/2016/6860573). Therefore, the discussion should be revised considering such points.
Author Response
Comments 1:
The article “Effects of Different Intensities of Endurance Training on 2 Neurotrophin Levels and Functional and Cognitive Outcomes in Post-Stroke Adults: A Randomised Clinical Trial” by Gorna and coworkers reported the data obtained on selected blood neurotrophins and physical and cognitive outcomes in post-ischemic stroke patients after different intensities of endurance training combined with standard neurorehabilitation.
Two different physical rehabilitation programmes were assigned to patients: moderate-intensity continuous training (MICT) on a bicycle ergometer and standard neurorehabilitation or low-intensity continuous training (LICT) on a bicycle ergometer and standard neurorehabilitation.
The authors selected five common neurotrophins as markers of recovery and rehabilitation such as BDNF, GDNF, VEGF-A, IGF-23 and irisin, all regulating a multitude of neuronal functions, including neuron metabolism and neuroinflammation, and muscle metabolism. Blood levels of these neurotrophins were statistically correlated with cognitive function and physical performance. Interestingly, the authors found that after 3 weeks, MICT improved patients’ aerobic capacity and motor function more than LICT. The clinical beneficial effects observed after MICT are accompanied by changes in BDNF levels in peripheral blood.
In this regard, the authors suggest that determining the neurotrophins with a beneficial effect on neurorehabilitation could contribute to a personalized therapy programme aimed at improving motor activity after ischemic stroke.
Overall, the paper is interesting and well-arranged. I appreciate the clarity of the tables. However, some edits should be done:
Response 1:
Dear Reviewer
Thank you for providing these insights. We wish to express our sincerest appreciation for your insightful comments on our paper, which have helped us significantly improve the quality of our paper. Below, we address each comment and indicate the location of changes, which are marked in yellow, in the revised manuscript.
Comments 2: Line 4 of the manuscript. Authors should specify that patients are affected by ischaemic stroke and should add the word “ischaemic” in the title as follows “...and cognitive outcomes in post-ischemic stroke adults…”
Response 2: Thank you for your suggestion. In the revised manuscript, we have added the word “ischaemic”(Line 4).
Comments 3: The introduction is long and redundant. In my opinion, the authors should resume the mechanism of action of the neurotrophins in a few sentences. For this reason, the introduction should be rewritten accordingly.
Response 3: We appreciate your suggestion to resume the mechanism of action of the neurotrophins in a few sentences. This fragment has been removed from introduction: ‘’Neurotrophins have an important role in nervous system plasticity in post-stroke survivors [5,6]. Therefore, knowledge of the physiological and clinical significance of neurotrophins may allow the optimisation of individual goals of motor and functional rehabilitation among patients with various neurological dysfunctions. Aerobic exercise may enhance neuroplasticity and alter the levels of synaptic proteins and neurotrophins [7,8]. Participation in physical activity causes higher cerebral oxygen supply and perfusion that is linked to enhancing neurogenesis, angiogenesis, and synaptic plasticity in the post-IS brain [9].’’
Comments 4: I warmly suggest adding a cartoon in which the mechanism of action of the neurotrophins chosen as markers could be schematically explained.
Response 4: Thank you for your suggestion. However, due to the length of the article (more than 20 pages) we decided not to add an additional cartoon.
Comments 5: Lines 209-212. The sentences are redundant and should be removed. The discussion should begin with: “The American Stroke Association….”
Response 5: Thank you for your suggestion. In the revised manuscript, we have removed the sentences: ‘’Authors should discuss the results and how they can be interpreted from the perspective of previous studies and of the working hypotheses. The findings and their implications should be discussed in the broadest context possible. Future research directions may also be highlighted.’’
Comments 6: Paragraph titled “Limitation and Implication of the study”: In this paragraph, authors reported the hypothesis that gut microbiota may interfere with rehabilitation intervention and the levels of neurotrophins. This observation is fascinating but falls outside the theme of the present manuscript. For this reason, it should be removed.
Response 6: Thank you for your suggestion. In the revised manuscript, we have removed the above -mentioned hypothesis from the paragraph titled “Limitation and Implication of the study”.
Comments 7: Line 578. The word “determined” should be replaced by the word “disclose”.
Response 7: Thank you for your suggestion. In the revised manuscript, we have replaced the word “determined” by the word “disclose”.
Comments 8: Table 3: Authors should indicate the p-value for differences between the two experimental groups for each marker.
Response 8: In our manuscript we have not indicated the p - values because the p - values give no information about the size or direction of an effect. ‘International Society of Physiotherapy Journal Editors (ISPJE) recommended the use of estimation methods instead of null hypothesis statistical tests. The American Statistical Association Statement on P-Values and Statistical Significance stopped just short of recommending that declarations of ‘statistical significance’ be abandoned.’
Elkins MR, Pinto RZ, Verhagen A, Grygorowicz M, Söderlund A, Guemann M, Gómez-Conesa A, Blanton S, Brismée JM, Ardern C, Agarwal S, Jette A, Karstens S, Harms M, Verheyden G, Sheikh U. Statistical inference through estimation: recommendations from the International Society of Physiotherapy Journal Editors. J Physiother. 2022 Jan;68(1):1-4. doi: 10.1016/j.jphys.2021.12.001. Epub 2021 Dec 21. Erratum in: J Physiother. 2022 Apr;68(2):89. doi: 10.1016/j.jphys.2022.03.008.
Wasserstein, RL, Schirm, AL & Lazar, NA 2019, 'Moving to a World Beyond “p < 0.05”', American Statistician, vol. 73, no. sup1, pp. 1-19. doi: 10.1080/00031305.2019.1583913
Comments 9: The authors detected a decrease in the peripheral BDNF concentration in the MICT group compared with LICT as reported in the discussion paragraph. Contrariwise, several papers reported that the peripheral levels of BDNF increase after exercise (see for a review Lukkahatai et al., Biomedicines 2025, 13(2), 332; doi.org/10.3390/biomedicines13020332). Moreover, the authors should take into account the relationship between BDNF and the hypothalamic-pituitary-adrenal (HPA) axis that plays an important role in homeostasis maintenance (Tapia-Arancibia et al., Front Neuroendocrinol 2004; 25: 77–107). On the other hand, it is demonstrated that the adrenal gland represents a target of physical exercise (Brown et al., J Appl Physiol 2007 Dec;103(6):1979-85. doi: 10.1152/japplphysiol.00706.2007; Bartalucci et al., Histol Histopathol 2012 Jun;27(6):753-69.doi: 10.14670/HH-27.753; Hotting et al., Neural Plasticity Volume 2016, doi.org/10.1155/2016/6860573). Therefore, the discussion should be revised considering such points.
Response 9: In the revised manuscript, we have added in the discussion section this fragment ‘’In contrast, Lukkahatai et al. [41] in scoping review revealed that peripheral levels of BDNF increase after exercise. The inconsistent findings on BDNF changes may have been partly due to neuroendocrinological mechanisms such as hypothalamic-pituitary-adrenal axis or adrenal gland function [42,43]. The above-mentioned phenomenon ought to be clarified and further research is needed (Lines 331 - 335).
References:
- Lukkahatai, N.; Ong, I.L.; Benjasirisan, C.; Saligan, L.N. Brain-Derived Neurotrophic Factor (BDNF) as a Marker of Physical Exercise or Activity Effectiveness in Fatigue, Pain, Depression, and Sleep Disturbances: A Scoping Review. Biomedicines. 2025, 31;13(2):332. doi: 10.3390/biomedicines13020332
- Tapia-Arancibia, L.; Rage, F.; Givalois, L.; Arancibia, S. Physiology of BDNF: focus on hypothalamic function. Front Neuroendocrinol. 2004, 25(2):77-107. doi: 10.1016/j.yfrne.2004.04.001
- Brown, D.A.; Johnson, M.S.; Armstrong, C.J.; Lynch, J.M.; Caruso, N.M.; Ehlers, L.B.; Fleshner, M.; Spencer, R.L.; Moore, R.L. Short-term treadmill running in the rat: what kind of stressor is it? J. Appl. Physiol. (1985). 2007, 103(6):1979-85. doi: 10.1152/japplphysiol.00706.2007
Best regards,
Authors

Reviewer 2 Report
Comments and Suggestions for Authors
The overall research is good. The design is appropriate, the work is quite solid, and the results are pretty supportive.
Prior to the final publish, I would like to point out some minor suggestions and questions:
1. Is the study single-blind (participants blinded) or double-blind (investigators and participants blinded)?
2. About the sample size, I think it might be better to include more participants;
3. Did author conduct any pilot studies or referencing meta-analyses?
4. For all the participants involved, are they randomly assigned to groups?
5. It is usually good to have participants complete at least 80% sessions, but how do you track adherence beyond session attendance?
6. Does the authors have any plans for long-term investigations? For example, a 6-week and 3-month follow-ups to determine retention of benefits.
7. In the biomarker study, what is the environment (like temperature) expose to the participants?
8. In the biomarker study, what is the speed and time the centrifugation. And after that, did authors directly store in -80, or fridge it at 4, then freeze to -80?
Author Response
Comments 1: The overall research is good. The design is appropriate, the work is quite solid, and the results are pretty supportive. Prior to the final publish, I would like to point out some minor suggestions and questions:
Response 1:
Dear Reviewer
Thank you for providing these insights. We wish to express our sincerest appreciation for your insightful comments on our paper, which have helped us significantly improve the quality of our paper. Below, we address each comment and indicate the location of changes, which are marked in yellow, in the revised manuscript.
Comments 2: Is the study single-blind (participants blinded) or double-blind (investigators and participants blinded)?
Response 2: The study is single-blind (participants blinded).
Comments 3: About the sample size, I think it might be better to include more participants.
Response 3: Thank you for your suggestion. The study met the criteria for the minimum sample size of participants that would be appropriate to detect a significant minimal clinically important difference between the groups. In future studies we will try to establish cooperation with other hospitals in order to include more participants.
Comments 4: Did author conduct any pilot studies or referencing meta-analyses?
Response 4: We have conducted a current evidence and qualitative systematic review to check if there is any relationship between participating in endurance effort and the circulating BDNF concentration in adult post-stroke individuals. And to answer the questions: Are the recent clinical trials examining the influence of endurance exercise on BDNF expression, including their methodological approach, reliable?
Górna S, Domaszewska K. The Effect of Endurance Training on Serum BDNF Levels in the Chronic Post-Stroke Phase: Current Evidence and Qualitative Systematic Review. J Clin Med. 2022 Jun 20;11(12):3556. doi: 10.3390/jcm11123556.
Comments 5: For all the participants involved, are they randomly assigned to groups?
Response 5: All participants were randomly assigned to groups.
Comments 6: It is usually good to have participants complete at least 80% sessions, but how do you track adherence beyond session attendance?
Response 6: Throughout the duration of their participation in the project, patients stayed in the hospital in-ward, from which they were not allowed to leave. Any failure to attend a single rehabilitation session was noted in the patient's chart. During daily medical visits, patient charts and possible absences from rehabilitation were discussed.
Comments 7: Does the authors have any plans for long-term investigations? For example, a 6-week and 3-month follow-ups to determine retention of benefits.
Response 7: Thank you for your suggestion. In the next studies we plan to evaluate the effect after 3- month follow-up and determine motor and functional retention of benefits.
Comments 8: In the biomarker study, what is the environment (like temperature) expose to the participants?
Response 8: All patients participating in the study were exposed to a similar environment. During rehabilitation and rest, the rooms were kept at room temperature (20 - 22 0C).
Comments 9: In the biomarker study, what is the speed and time the centrifugation. And after that, did authors directly store in -80, or fridge it at 4, then freeze to -80?
Response 9: The blood was left to clot at room temperature for 30 minutes. The samples were centrifuged at 2000 rpm for 10 minutes at 4 0C, then aliquoted into microcentrifuge tubes and separated samples were frozen and kept at - 80 0C.
Best regards,
Authors

Reviewer 3 Report
Comments and Suggestions for Authors
The authors conducted an interesting study aimed at examining the effects of different intensities of endurance training combined with standard neurorehabilitation on selected blood biomarkers and physical outcomes in adult post-stroke individuals. Their results, derived from patient data, indicated that after 3 weeks, an intervention combining MICT with standard neurorehabilitation was significantly more beneficial in improving aerobic capacity and arm motor function than an intervention combining LICT and standard neurorehabilitation.
Below are my comments:
-
Line 61: I suggest briefly introducing the biomarkers mentioned, explicitly stating their function. Most of these are not directly related to neuroplasticity (for example, IGF-1 is closely associated with the action of growth hormone and development during adolescence). While this information is provided later in the manuscript, it should also be specified in the initial sections.
-
In Table 2: A legend is needed to clarify the abbreviations used in the text; this comment also applies to the other tables.
-
Materials and Methods: The section is adequate, and the statistical analysis aligns well with the authors' objectives.
-
Lines 208-212: I believe there is a formatting error; please revise.
-
Line 239: Is there a specific reason why the authors chose 3 weeks as the cutoff? Is this the period of maximum neuronal plasticity following ischemic stroke?
-
Line 398: Based on my previous comment, what do the authors expect for longer-term follow-ups? Would the 6- to 12-month outcome continue to differ?
-
Language: Minor improvements to the English translation are needed.
Minor improvements to the English translation are needed.
Author Response
Comments 1: The authors conducted an interesting study aimed at examining the effects of different intensities of endurance training combined with standard neurorehabilitation on selected blood biomarkers and physical outcomes in adult post-stroke individuals. Their results, derived from patient data, indicated that after 3 weeks, an intervention combining MICT with standard neurorehabilitation was significantly more beneficial in improving aerobic capacity and arm motor function than an intervention combining LICT and standard neurorehabilitation. Below are my comments:
Response 1:
Dear Reviewer
Thank you for providing these insights. We wish to express our sincerest appreciation for your insightful comments on our paper, which have helped us significantly improve the quality of our paper. Below, we address each comment and indicate the location of changes, which are marked in yellow, in the revised manuscript.
Comments 2: Line 61: I suggest briefly introducing the biomarkers mentioned, explicitly stating their function. Most of these are not directly related to neuroplasticity (for example, IGF-1 is closely associated with the action of growth hormone and development during adolescence). While this information is provided later in the manuscript, it should also be specified in the initial sections.
Response 2: Thank you for your suggestion. However, other reviewer suggest that the introduction is long and redundant and we should resume the mechanism of action of the neurotrophins in a few sentences. For this reason we removed this paragraph ‘’Neurotrophins have an important role in nervous system plasticity in post-stroke survivors [5,6]. Therefore, knowledge of the physiological and clinical significance of neurotrophins may allow the optimisation of individual goals of motor and functional rehabilitation among patients with various neurological dysfunctions. Aerobic exercise may enhance neuroplasticity and alter the levels of synaptic proteins and neurotrophins [7,8]. Participation in physical activity causes higher cerebral oxygen supply and perfusion that is linked to enhancing neurogenesis, angiogenesis, and synaptic plasticity in the post-IS brain [9].’’
Comments 3: In Table 2: A legend is needed to clarify the abbreviations used in the text; this comment also applies to the other tables.
Response 3: Thank you for your valuable feedback. As suggested, we have included abbreviations under each table.
Comments 4: Materials and Methods: The section is adequate, and the statistical analysis aligns well with the authors' objectives.
Response 4: Thank you for your positive comment.
Comments 5: Lines 208-212: I believe there is a formatting error; please revise.
Response 5: Thank you for noticing a formatting error. We have corrected it.
Comments 6: Line 239: Is there a specific reason why the authors chose 3 weeks as the cutoff? Is this the period of maximum neuronal plasticity following ischemic stroke?
Response 6: Neuroplasticity impacts physical recovery and is mostly generated during the early six-month post-stroke before achieving plateau. The quantitative synthesis of systematic review of 21 parallel RCTs showed that mean duration of rehabilitation was two weeks to six months after stroke.
Due to the lack of established recommendations regarding the duration of rehabilitation in the subacute period of stroke, we decided to assess whether a 3-week comprehensive rehabilitation can change motor and cognitive functions and the concentration of selected neurotrophins.
Rahayu UB, Wibowo S, Setyopranoto I, Hibatullah Romli M. Effectiveness of physiotherapy interventions in brain plasticity, balance and functional ability in stroke survivors: A randomized controlled trial. NeuroRehabilitation. 2020;47(4):463-470. doi: 10.3233/NRE-203210.
Clark B, Whitall J, Kwakkel G, Mehrholz J, Ewings S, Burridge J. The effect of time spent in rehabilitation on activity limitation and impairment after stroke. Cochrane Database Syst Rev. 2021 Oct 25;10(10):CD012612. doi: 10.1002/14651858.
Comments 7: Line 398: Based on my previous comment, what do the authors expect for longer-term follow-ups? Would the 6- to 12-month outcome continue to differ?
Response 7: There is currently insufficient evidence to recommend a minimum beneficial daily and total amount in clinical practice. Neuroplasticity impacts physical recovery and is mostly generated during the early six-month post-stroke before achieving plateau, so in longer-term follow-ups we can expect the maintenance or slight improvement of previously achieved rehabilitation effects in motor and cognitive function and neurotrophin levels.
Clark B, Whitall J, Kwakkel G, Mehrholz J, Ewings S, Burridge J. The effect of time spent in rehabilitation on activity limitation and impairment after stroke. Cochrane Database Syst Rev. 2021 Oct 25;10(10):CD012612. doi: 10.1002/14651858.
Comments 8: Language: Minor improvements to the English translation are needed.
Response 8: Before submission to the IJMS the manuscript has been sent to Proof-Reading-Service.com for editing and proofreading .
Best regards,
Authors

Round 2
Reviewer 1 Report
Comments and Suggestions for Authors
The authors have addressed all the concerns raised by the reviewer. Therefore, the manuscript can be accepted in its present form.
Reviewer 3 Report
Comments and Suggestions for Authors
No more comments.